# Familial 4p Interstitial Deletion Provides New Insights and Candidate Genes Underlying This Rare Condition

**DOI:** 10.3390/genes14030635

**Published:** 2023-03-03

**Authors:** Jing Di, Leonard Yenwongfai, Hillary T. Rieger, Shulin Zhang, Sainan Wei

**Affiliations:** 1Department of Pathology and Laboratory Medicine, University of Kentucky, Lexington, KY 405362, USA; 2Department of Pediatric Genetics & Metabolism, Kentucky Children’s Hospital, Lexington, KY 405036, USA

**Keywords:** chromosome 4p, familial, interstitial deletion, SNP array analysis, cytogenetics

## Abstract

Chromosome 4p deletions can lead to two distinct phenotypic outcomes: Wolf-–Hirschhorn syndrome (a terminal deletion at 4p16.3) and less frequently reported proximal interstitial deletions (4p11-p16). Proximal 4p interstitial deletions can result in mild to moderate intellectual disability, facial dysmorphisms, and a tall thin body habitus. To date, only 35 cases of proximal 4p interstitial deletions have been reported, and only two of these cases have been familial. The critical region for this syndrome has been narrowed down to 4p15.33-15.2, but the underlying causative genes remain unclear. In this study, we report the case of a 3-year-old female with failure to thrive, developmental and motor delays, and morphological features. The mother also had a 4p15.2-p14 deletion, and the proband was found to have a 13.4-Mb 4p15.2-p14 deletion by chromosome microarray analysis. The deleted region encompasses 16 genes, five of which have a high likelihood of contributing to the phenotype: *PPARGC1A*, *DHX15*, *RBPJ*, *STIM2*, and *PCDH7*. These findings suggest that multiple genes are involved in this rare proximal 4p interstitial deletion syndrome. This case highlights the need for healthcare providers to be aware of proximal 4p interstitial deletions and the potential phenotypic manifestations.

## 1. Introduction

Two distinct phenotypes have been associated with deletions on chromosome 4p: Wolf-Hirschhorn syndrome (WHS; OMIM 194190)—terminal deletion [1], which is characterized by significant growth and severe intellectual disability, seizures, specific facial deformities (Greek warrior helmet appearance), and multiple congenital anomalies; and proximal interstitial deletions (4p11-p16) which have been rarely reported and may have distinct mild to moderate intellectual disability, multiple dysmorphic features including long face, up-slanted palpebral fissures with epicanthal folds, tall thin body habitus, and hyperextensible joints [2,3,4,5,6,7,8,9,10,11,12,13,14,15]. The phenotypic manifestation of proximal 4p deletion syndrome is generally less severe than that of WHS but is not well-known among healthcare professionals and confers a broad spectrum of congenital abnormalities. It may thus go undiagnosed until the affected person is older. Upon literature review, 35 cases with interstitial deletions of 4p have been described, with ages from prenatal to 77-year-old (mean 15.1, median 30), while males and females were equally distributed. To date, only two familial cases have been reported, and most of the reported deletions were detected by conventional chromosome analyses [3,7]. In addition, the critical region for proximal 4p interstitial deletion syndrome has been localized to 4p15.33-15.2, but the causative gene or genes remain elusive. In this study, we report a rare familial proximal interstitial 4p deletion case with an assessment of the pertinent deleted genes and a thorough literature review of similar deletions. We hope this report will help make this syndrome more recognizable among clinicians and provide a foundation for identifying the primary causative gene (or genes).

## 2. Case Presentation

A 3-year-old female, referred to as the proband, was brought to the Clinical Genetic Laboratories at the University of Kentucky for examination and testing. She was experiencing growth failure, delays in her physical and intellectual development, and had physical features that were of unknown origin. The proband appeared thin, had a small head with normal hair, larger ears, long eyelashes, upward-slanting eye openings, a low nasal bridge with broad nostrils, and a heart-shaped upper lip with a grooved philtrum, and a slightly enlarged tongue.

The proband’s mother, at the age of 23, had an intellectual disability and physical abnormalities including large knuckles, a tall and slender physique, a heart murmur, 50% loss of hearing in the left ear, vision abnormalities, and facial abnormalities. Her karyotype was previously tested when she was 9 years old and showed a deletion in the 4p15.2-p14 region. She was evaluated again at 10 years and 10 months and was found to have borderline microcephaly, anxiety, and frequent laughing, a flat mid-face, upward-slanting eye openings, folds over the inner corners of her eyes, borderline wide-set eyes, an upturned nasal tip, a narrow palate, a right crease, crowded teeth, bifid uvula, hypermobility of elbows, and developmental delays. She was born full-term and without any complications, weighing 6 lb 5 oz and measuring 19 inches long. Genetic testing was performed due to her distinctive physical features.

The proband also had two maternal uncles, ages 16 and 12, who were healthy and two maternal aunts, ages 19 and 5, who were also healthy, but their karyotypes were unavailable. The karyotype of the maternal grandmother was reported as normal. Multiple family members on her maternal grandfather’s side are said to have a similar appearance to the proband’s mother and her. The proband’s father was 29 years old and healthy. Both parents are Caucasian and are not known to be consanguineous.

## 3. Methods and Results

### 3.1. Cytogenetic Analysis

Using standard procedures, routine G-banded karyotyping was performed on cultured peripheral blood lymphocytes from the patient and her mother, at the 550-band level according to the International System for Human Cytogenomic Nomenclature (ISCN) 2020. Conventional chromosome studies on this proband showed a deletion of 4p15.2-p14 (Figure 1), the same as her mother.

### 3.2. High-Resolution SNP Array Analysis

Genomic DNA was isolated from peripheral blood lymphocytes with the QIAamp DNA Blood Mini kit (Qiagen, Inc., Valencia, CA, USA) according to the manufacturer’s instructions. DNA sample (250 ng) of the proband was hybridized to CytoScan HD arrays on an Affymetrix SNP array platform (Affymetrix; Thermo Fisher Scientific, Inc., Waltham, MA, USA). The CytoScan HD array contains more than 2.6 million markers for the copy number analysis. Of these markers, 1,950,000 are unique, non-polymorphic oligonucleotide probes, and 750,000 are SNP probes used for genotyping. The average marker spacing is one probe per 1.1 kb, with a mean spacing of one probe per 1.7 kb on non-gene backbones and one probe per 880 bp in intragenic regions. The ChAS 4.2.0.80 software (Affymetrix; Thermo Fisher Scientific, Inc) was used for copy number variants analyses. A 13.4-Mb deletion at 4p15.2-p14 encompassing 16 genes was identified on the proband (Table 1; Figure 2).

The clinical interpretation of this deletion was based on a scoring system recommended by the ACMG (American College of Medical Genetics) guidelines published in 2019 [16]. The available evidence in the literature is not enough to classify this finding as a pathogenic variant. However, the relatively consistent clinical manifestation pattern in many of the 35 previous cases (plus the two described here), and the similar genomic locations on 4p, suggest the possibility that this may be a recognizable genetic syndrome. Our molecular analyses show 16 genes in this deleted region for our proband, and five of these 16 genes, *PPARGC1A, DHX15, RBPJ, STIM2* and *PCDH7*, have a pLI score of more than 0.98, and thus likely to have phenotypic effects. These molecular findings combined with the prior clinical and cytogenetic studies may strengthen the case for defining a new proximal 4p interstitial deletion syndrome that is distinct from the well-established 4p16.3 (WHS) terminal deletion syndrome. 

## 4. Discussion

Proximal interstitial deletions of chromosome 4p are relatively rare. Upon careful literature review, fewer than 40 patients have been described (Appendix A). The age range is from prenatal to 77-year-old (mean 15.1, median 30), and females and males are equally distributed. The main clinical features of the reported cases including the presented cases are mild to moderate intellectual disability (96%), while over half the patients have multiple minor dysmorphic features including a long face, midface hypoplasia, upslanted fissures, epicanthal folds, large beaked nose, thick lower lip, high palate, and tall-thin body habitus. In addition, more than half of male patients have cryptorchidism and about 40% of patients show congenital heart diseases, including pulmonary stenosis, atrial septal defect, and Tetralogy of Fallot. Compared to “significant growth and severe intellectual disability, seizures, specific facial deformities (Greek warrior helmet appearance)” [1], manifestations of WHS, the phenotype of proximal 4p interstitial deletion syndrome is usually relatively mild and grants a broad spectrum of congenital abnormalities, however, it is not well recognized among health care providers. Therefore, it may be overlooked till the patient is older [17].

The critical region for proximal 4p deletion syndrome has been localized to 4p15.33-15.2, which is also distinct from the terminal location of WHS. Since most studies used traditional cytogenetic analysis, the causative gene or genes remain unknown. In this study, we found 16 genes in the deletion region, five of which are more likely to have phenotypic effects due to haploinsufficiency (pLI scores > 0.98). One or more of these five genes are likely to be the primary causative factors in the clinical manifestations. To our knowledge, there have been only two reported instances of familial interstitial 4p deletion, both of which were identified by conventional chromosome analyses [3,7]. A more detailed illustration of the genes implicated in the deleted region would improve our understanding of the correlation between genotype and phenotype. Recent reports have identified de novo interstitial 4p deletions using CMA and conventional G-banding cytogenetics similar to the methods used in our study. Mitroi et al. [13] and Park et al. [17] identified de novo 4p interstitial deletions in a proband with intellectual disability, mild dysmorphic features, and mild hypotonia, similar to our case. The authors speculated that SLIT2, KCNIP4, RBPJ, and LG12 genes are involved in neurologic development and skeletal abnormalities. The genes identified in our study significantly overlap with previous studies, however, we did not identify *SLIT2* and *KCNIP4* genes in the deleted 4p region. The variation in the methodology employed across different studies may, to some extent, account for the discrepancy.

Recombination signal-binding protein for immunoglobulin Kappa J region (RBPJ) encoded by *RBPJ* gene, is a transcriptional regulator playing a critical role in the Notch signaling pathway [18]. As described by Giaimo et al. [19], RBPJ can behave as a “molecular switch” from activation to repression Notch signal transduction cascades during blood cell development. The on/off switch of Notch signaling is well known to regulate cell proliferation and apoptosis in multiple organs [20]. Previous studies have also shown that the reduced expression of Notch pathway proteins can lead to multiple developmental defects [21,22], suggesting that Notch pathway is likely dosage dependent. Haploinsufficiency of Notch signaling components including *RBPJ* gene has been reported in families with Adams-Oliver syndrome (AOS [MIM 100300]) [23], which usually shows autosomal-dominant inheritance, with the most common features being terminal limb malformations and congenital cutis aplasia [23]. The presented proband has haploinsufficiency of *RBPJ* and shows mainly facial deformations, long/thin body status, and her mother had congenital cardiac disease but didn’t show severe vascular and limb defects of AOS, therefore, other components of the pathway must have played a role. For example, RBPJ-mediated Notch signaling strength has been shown to be involved in mesenchymal cell proliferation and skeletal formation [24], epidermis and hair follicle development [25], and vascular structure formation [26]. Moreover, Krebs et al. showed that *RBPJ* conditional-knockout mice have arteriovenous malformations [18]. The disruption of RBPJ by a translocation was also reported in an individual who presented with a similar phenotype to proximal 4p deletion [27].

The stromal interaction molecule STIM1 and STIM2 function as sensors of Ca^2+^ in the endoplasmic reticulum and maintain high cytosolic calcium levels by regulating Ca^2+^ influx [28]. Altered expressions of *STIM1* in mouse and cell models have demonstrated its critical function in hypertension, adipocyte differentiation, cognitive impairment, and cancer [29,30,31]. Unlike STIM1, the function of STIM2 has not been fully explored. STIM1-deficient mice have been shown perinatally lethal, while STIM2-deficient mice show a growth delay and begin to die around 4–8 weeks after birth [31,32,33]. STIM2 seems to protect hippocampal neurons from amyloid synaptotoxicity [34], and inhibition of *STIM2* by miR-128 causing memory impairment in Alzheimer’s disease mouse model [35], which suggests the neuroprotective potentials of STIM2 [36]. Recent studies also show that STIM2 involves the invasion and metastasis of breast cancer [37] and cancer-induced inflammation in leukemia [38]. The DEAH-box RNA helicase 15 (DHX15) is implicated in RNA metabolism, including splicing and ribosome biogenesis [39,40]. It’s also involved in several biological processes, such as regulating antiviral innate immune responses in enterocytes [41] which is recently revealed crystal structure by Murakami et al. [42]. Many studies have discussed the involvement of DHX15 in several cancer progression, either enhancing or suppressing it [43,44,45]. More importantly, *DHX15* gene knockdown in Jurkat, raji and NB4 cells by Chen et al. and Pan et al. groups have shown to induce cell apoptosis, arrest cell cycle, and inhibit cell proliferation [43,46]. The haploinsufficiency of *DHX15* in our proband could imply the involvement of DHX15 in the facial dysmorphic features and developmental delay.

Peroxisome proliferator-activated receptor γ coactivator-1 α (PGC-1α/PPARGC1A) is a crucial transcriptional coactivator required for lipid metabolism in the mitochondria [47], implicated in the pathogenesis of neurodegenerative disorders [48]. In addition, the role of PGC-1α in the pathogenesis of age-related macular degeneration is related to direct or indirect interactions with vascular endothelial growth factor (VEGF), telomerase, oxidative stress, and autophagy-related mechanisms [49]. PGC-1α has unique roles in the central nervous system of mouse models, such as in the maturation of neuronal cells and synaptogenesis, while deficiencies promote behavioral dysfunction by impacting the inhibitory signals of dopaminergic and interneurons [50]. *PCDH7* encodes the cadherin protein of brain–heart protocadherin (BHPCDH), which mediates calcium-dependent cell-cell adhesion [51]. *PCDH* genes are essential for brain development [52]. PCDH7 neural fold protocadherin (NFPC) has been shown to regulate differentiation of the embryonic ectoderm [53], neural tube formation [54], and cell morphology [55]. *PCDH7* promoter activities have been observed to be down-regulated by MECP2, and mutations of the *MECP2* gene cause MECP2 reduction and BHPCDH up-regulation in postmortem brains from Rett syndrome patients [56]. These findings suggest that haploinsufficiency of *PCDH7* may be involved in the mental developmental delay of our proband and likely other proximal 4p deletion syndrome patients. However, the understanding of the molecular mechanisms of PCDH7 is still poor and requires further study.

## 5. Conclusions

We reviewed the phenotypic abnormalities of previously reported patients with similar proximal 4p interstitial deletions and discussed the gene functions deleted within this region. This report contributes to expanded knowledge of the genetic background of proximal 4p interstitial deletion syndrome and facilitates accurate genetic counseling, including recurrence risks and prenatal testing options. The genetic investigations performed in the study have allowed us to define better the critical chromosomal region and the best candidate genes for follow-up studies. We anticipate this is an emerging distinct genetic syndrome that is proximal to the well-described 4p16.3 (WHS) terminal deletion syndrome.

## Figures and Tables

**Figure 1 genes-14-00635-f001:**
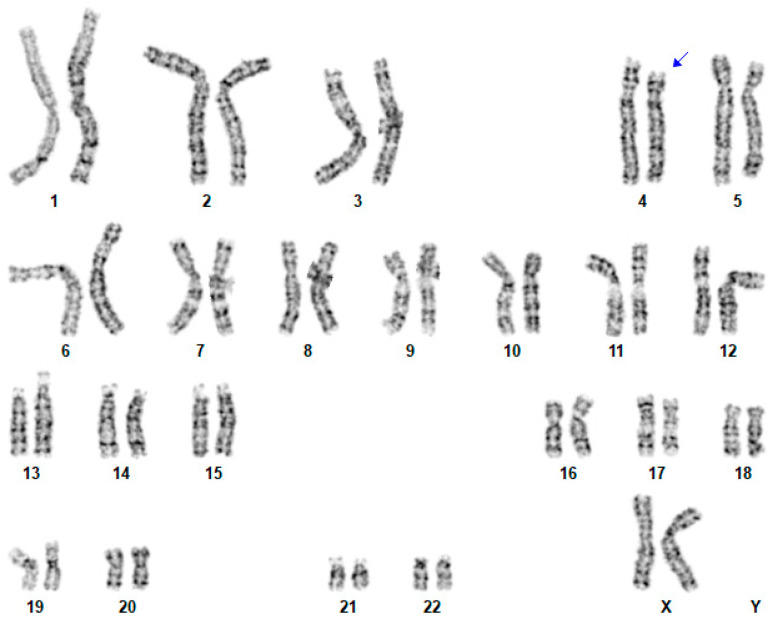
Conventional cytogenetic analysis on this proband. A karyotype 46,XX,del(4)(p15.2-p14). Blue arrow indicates where the proximal deletion is located.

**Figure 2 genes-14-00635-f002:**
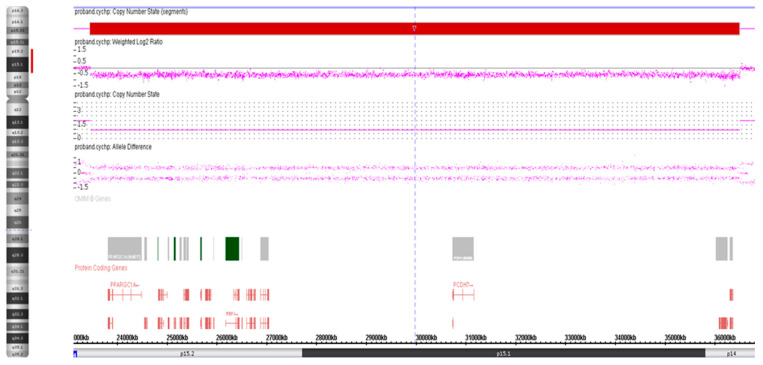
Whole-genome CMA analysis on uncultured peripheral blood lymphocytes of the proband. An 13.4-Mb deletion at 4p15.2-p14 [ISCN arr[GRCh37] 4p15.2-p14(23443552_36483601)×1].

**Table 1 genes-14-00635-t001:** Genes deleted in the 4p15.2-14 region in the presented case.

Gene	OMIM	Protein/Transcript Name	Function/Dysfunction of Gene Product
*PPARGC1A*	604,517	Peroxisome proliferative activated receptor γ	Transcriptional coactivator for steroid receptors and nuclear receptors
*DHX15*	603,403	DEAH box polypeptide 15	Nuclear ATP-dependent helicase
*SOD3*	185,490	Superoxide dismutase 3	Free radical detoxification
*LGI2*	608,301	Leucine-rich glioma inactivated protein 2	May be involved in axonal path finding
*SEPSECS*	613,009	O-phosphoserin tRNA-selenocystein tRNA synthase	Pontocerebellar hypoplasia type 2D (AR)
*PI4K2B*	612,101	Phosphatidylinositol 4-kinase type 2 β	Phosphatidylinositol 4-kinase type 2 β
*ZCCHC4*	611,792	Zinc finger CCHC domain-containing protein 4	May be a methyltransferase
*ANAPC4*	606,947	Anaphase-promoting complex subunit 4	Component of anaphase-promoting complex/cyclosome, a cell cycle-regulated E3 ubiquitin ligase and the G1 phase of the cell cycle
*SLC34A2*	604,217	Sollute carrier family 34 (sodium, phosphate cotransporter) member 2	Testicular microlithiasis. Pulmonary alveolar microlithiasisy (AR)
** *RBPJ* **	147,183	Recombination signal-binding protein for kappa J region	Adam-Oliver syndrome 3 (AD)
*CCKAR*	118,444	Colecystokinin A receptor	Receptor for cholecystokinin with role in colecystokinin induced regulation of satiety
*PCDH7*	602,988	Procadherin 7 isoform c precursor	Mediation of calcium dependent cell-cell adhesion expressed predominantly in SNC
*STIM2*	610,841	Stromal interaction molecule 2	Regulation of basal cytosolic and endoplasmic reticulum Ca^2+^ concentrations
*SMIM20*	617,465	Small Integral Membrane Protein 20	Component of the MITRAC complex that regulates cytochrome C oxidase assembly
*ARAP2*	606,645	Centaurin, Delta-1	Phosphatidylinositol 3,4,5-trisphosphate-dependent GTPase-activating protein that modulates actin cytoskeleton remodeling by regulating ARF and RHO family members
*DTHD1*	616,979	Death Domain-containing protein 1	Death domain containing proteins function in signaling pathways and formation of signaling complexes, as well as the apoptosis pathway

## Data Availability

All data that supports the findings of this study are included in this published article and its Appendix A.

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
