# Peer review of "Familial 4p Interstitial Deletion Provides New Insights and Candidate Genes Underlying This Rare Condition"

_genes, 2023, doi:10.3390/genes14030635_

Round 1
Reviewer 1 Report
This manuscript is well-written and highly clinically relevant. It provided important information and added our understanding for the 4p interstitial deletion associated conditions. There are a few minor points to be addressed:
1. I understand that for this rare condition, most of published papers only have conventional cytogenetics done, but for those did have microarray results, it will be beneficial to compare the findings especially genes involved with the current study and do genotype-phenotype correlation, if possible, to highlight the critical region/genes. At least, give some discussion addressing such previous publications.
2. Need to be carefully check ISCN for karyotype throughout the manuscript, to be consistent whether is by ISCN2016 or 2020. If 4p banding is from pter to centromere, it is following ISCN2020.
Author Response
Dear reviewer,
Thank you for your valuable feedback. We have taken your comments into consideration and made the following changes:
This manuscript is well-written and highly clinically relevant. It provided important information and added our understanding for the 4p interstitial deletion associated conditions. There are a few minor points to be addressed:
1. I understand that for this rare condition, most of published papers only have conventional cytogenetics done, but for those did have microarray results, it will be beneficial to compare the findings especially genes involved with the current study and do genotype-phenotype correlation, if possible, to highlight the critical region/genes. At least, give some discussion addressing such previous publications.
Response 1: Thank you for the comments! We have discussed and addressed the correlations with previous publications between line 147 to 158: To our knowledge, there have been only two reported instances of familial interstitial 4p deletion, both of which were identified by conventional chromosome analyses[3,7]. A more detailed illustration of the genes implicated in the deleted region would improve our understanding of the correlation between genotype and phenotype. Recent reports have identified de novo interstitial 4p deletions using CMA and conventional G-banding cytogenetics similar to the methods used in our study. Mitroi et al. [13]and Park et al. [17]identified de novo 4p interstitial deletions in a proband with intellectual disability, mild dysmorphic features, and mild hypotonia, similar to our case. The authors speculated that SLIT2, KCNIP4, RBPJ, and LG12 genes are involved in neurologic development and skeletal abnormalities. The genes identified in our study significantly overlap with previous studies, however, we did not identify SLIT2 and KCNIP4 genes in the deleted 4p region. The variation in methodology employed across different studies may, to some extent, account for the discrepancy."
We also summarized the comparisons in a table for your reference:
|
Paper |
Genes involved (CMA) |
Phenotype observed |
|
Jing et al -2023 |
PPARGC1A, DHX15, SOD3, LG12, SEPSECS, P14K2B, ZCCHC4, ANAPC4, SLC34A2, RBPJ, CCKAR, PCDH7, STIM2, SMIM20, ARAP2, DTHD1
Genes in yellow were the common genes found in both studies.
Method: 13.4 Mb deletion Affymetrix; Thermo Fisher Scientific, Inc., Waltham, MA, USA |
Thin and small head, large ears, long eye lashes, upward-slanting eye openings, low nasal bridge, broad nostrils, heart shaped upper lip, grooved philtrum, slightly enlarged tongue. |
|
Park et al-2020 Male De novo interstitial 4p deletion |
19 genes SLIT2, MIR218-1, KCNIP4 ADGRA3, GBA3, PPARGC1A, DHX15, SOD3, LGI2, SEPSECS, PI4K2B, ZCCHC4, ANAPC4, SLC34A2, SMIM20, RBPJ, CCKAR, STIM2, and PCDH7
Genes in red, highly expressed in brain tissue with potential roles in neurodevelopment.
Method: 12 Mb deletion CytoScan 750K chromosomal microarray (Affymetrix, Santa Clara, CA, USA) |
Tall, thin body, Hypotonia, mental retardation, hypertelorism |
|
Mitroi et al- 2017 De novo interstitial 4p deletion |
SLIT2, KCNIP4, MIR 218–1, GPR125, GBA3, PPARGC1A, DHX15, SOD3, LGI2, SEPSECS, P14K2B, ZCCHC4, ANAPC4, SLC34A2, RBPJ, CCKAR, PCDH7, STIM2,
Method used: 13.34 Mb Agilent Sure Print G3 Human Genome CGH+SNP, 4_180K, Microarray Kit |
Long face with a high forehead, deep-set eyes, puffy eyelids, broad andflat nasal bridge, lateralflaring of the nostrils, long philtrum, and a thick and proeminent lower lip His skin showed one café-au-lait spot. His ears consisted of proeminent and thick lobes, and they were very close to his head. He had pectus excavatum, broad hands and feet, and clinodactyly of the toes |
2. Need to be carefully check ISCN for karyotype throughout the manuscript, to be consistent whether is by ISCN2016 or 2020. If 4p banding is from pter to centromere, it is following ISCN2020.
Response 2: Thank you for the comments! This has been revised in compliance of ISCN 2020.

Reviewer 2 Report
I read with interest the manuscript. This case report is well written and informative.
I believe it deserves consideration for publication after a few minor revisions.
Please find here my comments:
All genes name should be italic.
Could authors offer more information regarding prenatal and neonatal history of the reported case and try to correlate with the genotype based on literature data?
Could authors offer information about cerebral MRI assessment of the reported case?
Rows 116, 122 - Use intellectual disability instead of mental retardation
Row 154 - Replace retardation with delay
Author Response
Dear reviewer,
Thank you for your valuable feedback. We have taken your comments into consideration and made the following changes:
I read with interest the manuscript. This case report is well written and informative.
I believe it deserves consideration for publication after a few minor revisions.
Please find here my comments:
1. All genes name should be italic.
Response 1: Thank you for bringing this to our attention! We have made the necessary updates and all gene names are now in italics.
2. Could authors offer more information regarding prenatal and neonatal history of the reported case and try to correlate with the genotype based on literature data?
Response 3: Thank you for your comment. However, the reported case was not born in our institution, and as a result, we are unable to obtain the prenatal and neonatal history of the reported case.
3. Could authors offer information about cerebral MRI assessment of the reported case?
Response 3: Thank you for your comment. However, we would like to clarify that there is no record of an MRI assessment for the reported case.
Rows 116, 122 - Use intellectual disability instead of mental retardation
Response: All mental retardation have been replaced with intellectual disability.
Row 154 - Replace retardation with delay
Response: Retardation has been replaced wtih delay.